# A Continuous Extraction Protocol for the Characterisation of a Sustainably Produced Natural Indigo Pigment

**DOI:** 10.3390/life14010059

**Published:** 2023-12-29

**Authors:** Elia Frignani, Veronica D’Eusanio, Mauro Grandi, Laura Pigani, Fabrizio Roncaglia

**Affiliations:** 1Department of Chemical and Geological Sciences, University of Modena and Reggio Emilia, 41125 Modena, Italy; veronica.deusanio@unimore.it (V.D.); laura.pigani@unimore.it (L.P.); 2National Interuniversity Consortium of Materials Science and Technology (INSTM), 50121 Firenze, Italy; 3G2B S.r.l., via Guareschi 25-27, 46010 Curtatone, Italy; maurograndi@iol.it

**Keywords:** indigo, bio-based pigment, solvent extraction, characterisation

## Abstract

The sustainable industrial production of indigo necessitates a unique extraction process to separate the plant-derived compounds. Calcium compounds are added to encourage hydrolysis of these precursors and to facilitate the isolation of the final form, resulting in an organic–inorganic composite pigment with unspecified characteristics. In this study, we devised a continuous solvent extraction procedure to fractionate the organic indigoid phase within the composite pigment. Overcoming challenges posed by limited solubility in the common organic solvents, this method allows for the analysis of individual fractions, significantly enhancing resolution. Comprehensive characterisation using spectroscopic analysis, thermogravimetry, and UHPLC-MS/MS revealed the potential for quantifying primary components of the natural pigment and distinct differentiation from the synthetic dye. This approach also holds promise for establishing robust manufacturing practices in the industrial production of natural indigo.

## 1. Introduction

Indigo has historically been a primary substance for dyeing textiles [1], traditionally derived from plants such as Indigofera tinctoria, Polygonum tinctorium, Nereum tinctorium, Isatis tinctoria [2], and Persicaria Tinctoria [3]. Natural indigo contains two primary components, indigotin (providing a blue colour) and indirubin (contributing to red-violet tones) [4], collectively imparting a distinctive shade to the pigment. Unlike other plant-derived pigments, indirubin and indigotin are not present in their final form within the plant. Instead, three glycoside precursors—namely, isatan A, isatan B, and indican—are present in the leaves, and indigo is formed from these in the extraction medium [5].

The typical extraction process involves steeping the aerial part of the plant in water for a specified time at room temperature. After the removal of solid plant residues, calcium hydroxide (Ca(OH)_2_) is added to facilitate the hydrolysis of the plant-derived precursors and the isolation of the pigment. During this stage, air bubbling serves two functions: providing the necessary oxidant (O_2_) and enhancing pigment precipitation by the simultaneous formation of colloidal CaCO_3_, derived from the interaction of Ca(OH)_2_ with atmospheric CO_2_. This inorganic colloid, with a high specific surface area, strongly interacts with the developing pigment, resulting in an ‘indigo on CaCO_3_’ composite that can be easily filtered, and that corresponds to the commercially accepted powder. It appears as a fine blue powder with higher proportions of indigotin.

After extraction, the hydrolytic cleavage of the glycosidic bond of isatans and indican releases indoxyl [6] (Figure 1), which constitutes the primary pigment precursor in solution. The alkaline medium was proposed to induce the formation of the reactive indoxyl enolate [7,8], which is oxidised to 3-oxo-indenoline, isatin, and an unstable indoxyl radical. These intermediates give rise to indigotin and indirubin through the interaction with a second molecule of indoxyl or through self-coupling, respectively.

Hence, the formation of the pigment is influenced by the specific natural source, including factors such as plant species, maturation, and growing parameters, as well as the conditions employed during the extraction process [9]. Consequently, the natural pigment exhibits varying proportions of indigotin and indirubin, along with minor characteristic compounds.

In contrast, synthetic indigo comprises solely indigotin and is highly cost-effective [10]. However, its production involves the use of toxic and environmentally hazardous substances, like aniline and hydrogen cyanide [11]. The likelihood of production residues or synthetic by-products contaminating the final compound is not remote, an occurrence frequently associated with allergic reactions to clothing dyed with synthetic pigments [12]. Moreover, textile dyes can significantly contaminate water bodies, both at dyeing plants and laundries, releasing organic chemicals that enter the food chain and contributing to toxicity at various levels [13].

Natural pigment production stands out for its remarkable sustainability, relying solely on water, air, aqueous HCl, and Ca(OH)_2_. Alternative methods aimed at creating indigo-like molecules, known as indigoids, emphasise sustainability through the use of microorganisms [14,15,16]. The enzymes within these biological species are able to catalyse the hydroxylation of indole, leading to the formation of indoxyl, which follows the same dimerisation mechanisms depicted in Figure 1, to afford indirubin and indigotin [17,18,19].

Indirubin and its analogues have recently been shown to exhibit different biological properties, including anticancer [20] as well as anti-inflammatory actions [21]. Therefore, the production of toxin-free indirubin and its fractionation represents a valuable option to access active pharmaceutical ingredients too.

These characteristics foster the development of a value chain involving farmers, pigment producers, and end-users, particularly dyers. There is a growing inclination among numerous textile manufacturers to embrace ‘green pigments’, especially in their high-value products, as part of a commitment to sustainable production practices.

However, a precise characterisation of industrial natural indigo remains elusive, partly owing to the limited solubility of the organic indigoid fraction within the composite pigment in commonly used organic solvents. This challenge has hindered comprehensive analysis and assessment. What is the actual loading of the natural pigment on the CaCO_3_? And what is the ratio between indigotin and indirubin? Is it possible separate the present indigoids? This work aims to address these questions by undertaking the fractionation and a comprehensive characterisation of industrially produced composite samples. The implementation of a continuous solvent extraction technique marked a significant advancement, allowing for a thorough extraction and distinct separation of the main pigment’s components. The employment of UV-visible and IR spectroscopies, along with thermogravimetric analysis and UHPLC-MS [22], provided us with a comprehensive understanding of the major composition of the natural pigment. This approach also allowed for a distinct differentiation between the natural pigment and its synthetic counterpart.

## 2. Materials and Methods

All reagents and solvents used are of commercial grade, sourced from Merck (Darmstadt, Germany) or Carlo Erba reagents (Milan, Italy), and used as received.

### 2.1. Industrial Indigo Preparation

Persicaria Tinctoria plants were grown in an open field and sourced by G2B S.r.l., Curtatone, Italy. The freshly harvested plants were mixed in a tank with water, with a plant:water weight ratio of 1:10. During this maceration process, both the extraction of the precursors and their hydrolysis occurs (Figure 1). The plants residues were filtered out, followed by the introduction of a steady stream of blowing air. Ca(OH)_2_ was gradually added until the pH reached 12. At this point, simultaneous precipitation of the pigment and CaCO_3_ started, resulting in the formation of a water-insoluble composite, within approximately 8 h. Removal of the upper surface layer, which contained low-density residues, was followed by the addition of a 6 M aqueous HCl solution to adjust the pH to 7 and eliminate excess Ca(OH)_2_. Filtration through a cloth allowed for the separation of the paste, subsequently dried and ground to obtain the final natural indigo composite, referred to as Product A.

### 2.2. Sequential Extraction of the Organic Fraction

To favour the dissolution of the pigment in organic solvents, a carbonate-free natural indigo powder was obtained by treating a portion of the Product A with 4 M aqueous HCl until pH 6.5. Dissolution of the inorganic portion allowed for the isolation of a solid organic residue, retained on a Gooch funnel (porosity = 4), which was washed with water and oven dried at 60 °C overnight. The obtained powder is herein referred to as Product B. The solubility of a sample of Product B (0.05 g) was assessed through five subsequent extractions with a selected organic solvent (1 mL) that was mixed (10 min), sonicated (10 min), and separated through centrifugation at 4000 rpm for 20 min. The solid residue remaining from each extraction was subjected to the following extraction step. The combined organic solutions were dried (rotavapor, mechanical pump) to obtain the soluble fraction as a solid powder. Several solvents were tested, such as methanol, ethyl acetate, acetone, acetonitrile, and DMSO. In the case of the high boiling point DMSO, water was added to precipitate the solute, which was isolated through centrifugation.

### 2.3. Continuous Extraction of the Organic Fraction

The limited solubility of the pigment in common organic solvents, evidenced during the sequential extraction experiments (Section 2.2), prompted us for the development of an improved extraction method, implemented through the modification of a Dean Stark trap (Figure 2A) [23]. The solvent, contained in a round bottom flask, was refluxed on a cartridge containing a sample of Product A (Figure 2B). Once the trap was filled, the pigment solution fell back into the round-bottom flask, resulting in a continuous, hot extraction. To allow the free passage of the solvent through the pigment sample, chromatography silica gel 60 powder was mixed with the composite pigment in a 4:1 weight ratio. The progress of extraction was assessed by weighting the amount of extracted indigoids (see later), collecting the solution in the flask at different times. After each collection, another volume of solvent was inserted in order to continue the process. A qualitative evaluation of the content of the fractions can be conducted through UV-Visible spectroscopy (see later).

A carbonate-free (whole) pigment powder (herein referred to as Product C) is obtained with no fractionation, that is, waiting for the complete extraction, collection of the solvent solution (in the flask and in the trap), removal of the solvent (rotavapor), and drying (oven, 60 °C, overnight). The residue after this extraction appears as a brownish-yellow powder (Figure 2C). An indirubin-rich powder (herein referred to as Product D) is obtained by collecting the solvent solution after the first extraction time, of which the exact value is solvent dependent (see later). Collecting the solvent solution in the following extraction time, allow the selective extraction of an indigotin-rich powder (herein referred to as Product E).

### 2.4. UV-VIS Spectra

UV-Visible spectra were recorded using a Perkin-Elmer Lambda 650 UV-Vis spectrometer (Perkin-Elmer, Waltham, MA, USA) in the spectral range from 900 to 350 nm using a dual-beam configuration and 0.5 cm quartz cuvettes.

### 2.5. IR Spectra

Samples were first oven dried at 60 °C overnight. IR spectra were acquired with Jasco FT-IR 4700 equipped with ATR adapter, in a spectral range of 4000 to 400 cm^−1^.

### 2.6. Thermogravimetric Analysis

Thermogravimetric analysis was performed using a Seiko SSC 5200 thermal analyser (Seiko Instrument Inc., Chiba, Japan). The samples were first ground in an agate mortar, then approximately 4 mg of the powder were inserted in a platinum crucible. The measurements were conducted from room temperature (25 °C) to 1050 °C, at 10 °C/min increments, in a helium atmosphere.

### 2.7. UHPLC-MS/MS Analysis

UHPLC analysis was acquired on samples obtained through the continuous extraction (see Section 2.3) with methanol. A mixture of water and acetonitrile served as the mobile phase, with a gradient of 35% to 100% acetonitrile in 10 min, at a flow rate of 0.3 mL/min, at 40 °C. The UHPLC separation was performed injecting 5 µL of sample, resulting in a total run time of 25 min. A Hypersil GOLD™ C18 column (100 × 2.1 mm ID, 1.9 µm ps; Thermo Fisher Scientific, Waltham, MA, USA) was employed. Both positive and negative masses were scanned with a mass resolution of 35,000 at 200 *m*/*z* for the first ionisation. The automatic gain control was set to 1 × 10^6^ and the maximum injection time to 243 ms. The full mass scan resolution was set to 17,500 at 200 *m*/*z* and the AGC to 2 × 10^5^ and the maximum injection time to 100 ms. The data obtained were processed with Freestyle (v. 1.8 SP 2) software (Thermo Fisher Scientific, Waltham, MA, USA).

## 3. Results and Discussion

### 3.1. Sequentiual Extraction

Indigo’s components, either natural or synthetic, show different solubility in various solvents, with indirubin generally more soluble than indigotin, and decreasing in the following order: DMSO >> acetone ~ ethyl acetate > methanol ~ acetonitrile.

UV-Visible absorption spectroscopy proved to be a simple and valuable technique for characterising indigo components. A first set of measurements was performed on solutions, prepared by mixing 0.05 g of either Product A or synthetic indigo with an organic solvent (5.0 mL) and removal of the insoluble part through a 0.45 µm nylon membrane. Differences between the natural Product A (Figure 3B) and the synthetic counterpart (Figure 3A) appeared evident.

With DMSO, which is able to dissolve most of the pigment, the absorption maximum of synthetic indigo (composed of indigotin only) is reached at 619 nm (Figure 3A), while two partially overlapped peaks are detected for the natural pigment, revealing the two main components indigotin (~615 nm) and indirubin (~572 nm). The considerable shift at lower wavelengths observed for all other solvents in the case of natural pigment (Figure 3B) reveals the limited solubility of pigment components [24], more pronounced for the less soluble indigotin.

The higher solubility in DMSO was exploited to evaluate the extraction of the main pigment components from a carbonate-free powder obtained with acid treatment. So, a sample of 0.05 g of Product B was sequentially extracted five times with DMSO (using 1.0 mL each time, see Section 2.2), and the partially dissolved fraction was assessed through UV-Visible spectroscopy. The solid residue after the fifth extraction still had a blue colour, indicating an incomplete extraction of indigotin. The progress of the extraction, showed in Figure 4, reveals an ongoing shift in the absorption maximum from 562 nm to 622 nm, from the first to the fifth extraction, indicating the greater solubility of indirubin compared to indigotin, even in DMSO.

Among those tested (Section 2.2), DMSO appears to be a solvent suitable to appreciably solubilise both indigotin and indirubin in a relatively small volume, suggesting the possible use of a spectrophotometric UV-Vis quantitation through a calibration curve, that represents a straightforward technique compared to known alternatives, such as HPLC [4,25].

### 3.2. Continuous Extraction

The unsatisfactory sequential extraction of the pigment components with most of the common organic solvents prompted us to consider a continuous-type extraction. By means of a modified Dean Stark trap (see Section 2.3), we succeeded in the extraction of pigment components directly from the industrial samples (Product A). After few hours of extraction, the remaining solid residue appeared as a brownish-yellow powder (Figure 2C), an indication of the exhaustive extraction of the indigoids from the inorganic support. Evidently, the employment of the solvent at the boiling point brings a useful increment of solubility of the pigment’s components.

When acetone was used as the solvent, the extraction of the pigment was almost complete within three hours, corresponding to approximately 6% of the mass of the loaded Product A. The progress of the extraction can be monitored by collecting the solvent in the flask at a certain time and weighting the mass of the indigoids resulting from solvent removal (see Section 2.3). Another volume of solvent was then inserted into the system in order to obtain the following fraction. Table 1 shows the relative mass percentages (where 100% corresponds to the complete extraction, that is, 6 wt% of the loaded Product A sample) obtained at different extraction times, with each of the two different solvents. 

Indirubin is mostly extracted in the first phase of the process and, when acetone is employed, is almost complete in 15 min, corresponding to around 20% in mass with respect to the total indigoids mass. After the first hour of processing, 42% of the organic mass was extracted, and it was completed within three hours.

UV-Vis spectroscopy (Figure 5A) reveals a strong absorption peak at 540 nm at the start of extraction, while after 15 min, a comparable signal relative to indigotin (absorption maximum at 598 nm) appeared. The greater solubility of indirubin resulted in an enhancement of the UV-Vis signal. Therefore, collecting the solvent solution at around this time allowed for the separation of a fraction mostly constituted by indirubin (Product D).

The lower dissolution ability of the pigment’s compounds by methanol is reflected by longer extraction times compared to acetone. The extraction of the indirubin fraction (Product D) occurs in around 60 min (black curve in Figure 5B, Table 1), while the complete extraction of indigoids from Product A requires four hours (Table 1). Collecting the methanolic solution coming from the continuous extraction from 60 to 240 min, the fractionation of mostly indigotin can be obtained (Product E, red curve in Figure 5B).

When a sample of synthetic indigo was subjected to the continuous extraction process, indigotin was detected (blue curve of Figure 5B) almost exclusively, according to what was previously observed (Figure 3A).

### 3.3. Trasmittance IR Spectra

The IR spectroscopy characterisation of solid samples of Product A, Product B, and synthetic indigo are shown in Figure 6.

Synthetic indigo, mainly consisting of one component, showed the most defined IR signals (Figure 6C), but most of them are recognisable in Product B too (Figure 6B). Despite this, natural indigo composite samples (Product A) showed the loss of most of the vibrational structure (Figure 6A), possibly due to the interaction with CaCO_3_. In particular, the intense stretching N-H band at 3263 cm^−1^ and the carbonyl band at 1624 cm^−1^ were not detectable, suggesting a strong interaction between these functional groups of the organic pigment and the calcium “support”.

### 3.4. Thermogravimetric Analysis

A first characterisation of the simpler synthetic indigo showed a sharp thermogravimetric analysis (TGA) decomposition event centred at around 367 °C (Figure 7A), and this was assigned to indigotin. The minor mass loss, observed at 500 °C, was related to the decomposition of an organic residue arising from indigotin in a non-oxidising atmosphere, possibly related to vitrification and/or graphitisation [26]. The TGA characterisation of the natural indigo composite (Product A) shows a similar trend (Figure 7B), at least until 600 °C. We assumed that both indigotin and indirubin share the same decomposition temperature at around 354 °C, with little influence of the inorganic support. Most of the mass loss of Product A occurred at 700 °C, compatible with the carbonate decomposition into CaO.

These data confirmed the previous evaluations, indicating that Product A samples are composed of around 6 wt% of organic indigoids and around 90 wt% of CaCO_3_.

A better TGA resolution on natural indigo samples could be obtained when the inorganic support was removed, either through acid dissolution (Product B) or following solvent extraction (Product C). Both methods proved to be highly effective in the removal of CaCO_3_, resulting in the disappearance of the relative TGA signal at 700 °C.

Small events at 63 °C and 135 °C in the case of Product B (Figure 8A) are likely due to residual water. Events occurring at temperatures (490 and 760 °C) greater than that of indigotin/indirubin main decomposition (~354 °C) are possibly linked to the decomposition of complexes between organic compounds and inorganic species, such as chloride anion (Cl^−^). Approximately 20% of the total sample mass of Product B cannot be degraded, suggesting the presence of some water-insoluble inorganic residues.

The TGA plot relative to Product C (Figure 8B) shows a main indigotin/indirubin degradation event split into three components: the most intense one, presumably due to indigotin (~368 °C), results partially associated with two minor events at 311 and 393 °C. The combined mass loss from these three peaks accounts for 68% of the total sample mass. Of the two, the relatively most intense signal (at around 311 °C) was assigned to the decomposition of indirubin.

This assignment was performed according to a further thermal evaluation of a sample of Product D, obtained by the fractionated continuous extraction of Product A. The TGA/DTA of this sample, shown in Figure 9, reveals a strong decomposition event at 320 °C (green curve) that aligns well to the earlier event of the thermal features of a sample of Product C (red curve). Low intensity events occurring at higher temperatures (430 °C and 500 °C) reflect the presence of minor components within Product D, which are co-extracted with indirubin.

What became evident by analysing the fine structure of Figure 9 is that the continuous extraction technique allowed for an enhancement of the thermal analysis resolution. Coupled together, the two techniques represent a versatile method able to differentiate between natural and synthetic pigment. Moreover, further refinement with curve deconvolution could allow for the reliable determination of the relative amounts of the main components.

### 3.5. UHPLC-MS Analysis

The UHPLC analysis is known to be a valuable tool for quantitative and qualitative characterisation of indigotin, indirubin, and their precursors within plants [22,27,28]. However, the characterisation of industrially produced natural indigo suffers from the difficulty of achieving a complete extraction of these compounds from the matrix. The continuous extraction technique, here adapted to the indigo pigment (Section 2.3), proved to be a seamless match with UHPLC analysis, especially when a compatible solvent such as methanol is employed.

The typical retention time for indigotin, under the employed instrument method, was assessed through the injection of a sample of synthetic indigo continuously extracted (0–240 min) with MeOH. A sharp signal around 3.9 min was detected, whose identity was provided by a positive molecular ion mass of 263.08 *m*/*z*, corresponding to the adduct indigotin-H^+^. The LC-MS analysis of Product D, that is, the natural pigment composite continuously extracted with MeOH (fraction 0–60 min), is shown in Figure 10A. The chromatogram shows a main peak at 4.56 min, assigned to indirubin, and a number of small signals related to other organic substances present in the natural pigment. The peak at 3.92 min can be related to a minor amount of indigotin that is hardly detectable in the pertinent UV-Visible spectra (Figure 5B). This assignation is made clear by analysing the positive molecular ion trace of 263.08 *m*/*z* (Figure 10B), revealing that indirubin (3.92 min of retention time) exhibits the same ionisation pattern of indigotin. These data also give evidence of the ability of the LC column to separate the main pigment components.

The LC-MS analysis of Product E, that is, the natural pigment composite continuously extracted with MeOH (fraction 60–240 min), is shown in Figure 10C. The chromatogram shows the presence of both indirubin (3.92 min) and indigotin (4.54 min). It also reveals the presence of a number of additional co-extracted compounds, similar to those present in Product D (Figure 10A), whose identity and relative amount could be a useful fingerprint of the natural pigment, related to both the production method and the natural source.

Figure 10D represents the same LC-MS acquisition (Product E) but expressed as the Single Ion Current (263.08 *m*/*z*), giving a precise mass identification of indigotin and indirubin.

Overall, the UHPLC-MS technique proved to work seamlessly with the continuous extraction technique and demonstrated a high sensitivity in the detection of even the minor-occurring compounds within the natural pigment samples.

## 4. Conclusions

The sustainable production of textiles with ‘green indigo pigments’ requires innovation at both the industrial production and quality control levels. More stringent quality standards are necessary if the produced indigoids are intended for pharmaceutical applications. A comprehensive chemical characterisation of the materials is essential to establish good manufacturing practices and to differentiate between naturally derived and synthetic indigoids. However, this analysis faces a significant challenge due to the low solubility of these compounds in most organic solvents.

In this study, a continuous extraction method was developed in order to extract and fractionate the organic compounds contained in industrially produced natural indigo composite samples. The combination of solvent extraction and analysis—using simple techniques like UV-Vis spectroscopy and TG, and/or more advanced methods like UHPLC-MS/MS—shows significant promise for the fully qualitative and quantitative characterisation of natural indigo production.

## Figures and Tables

**Figure 1 life-14-00059-f001:**
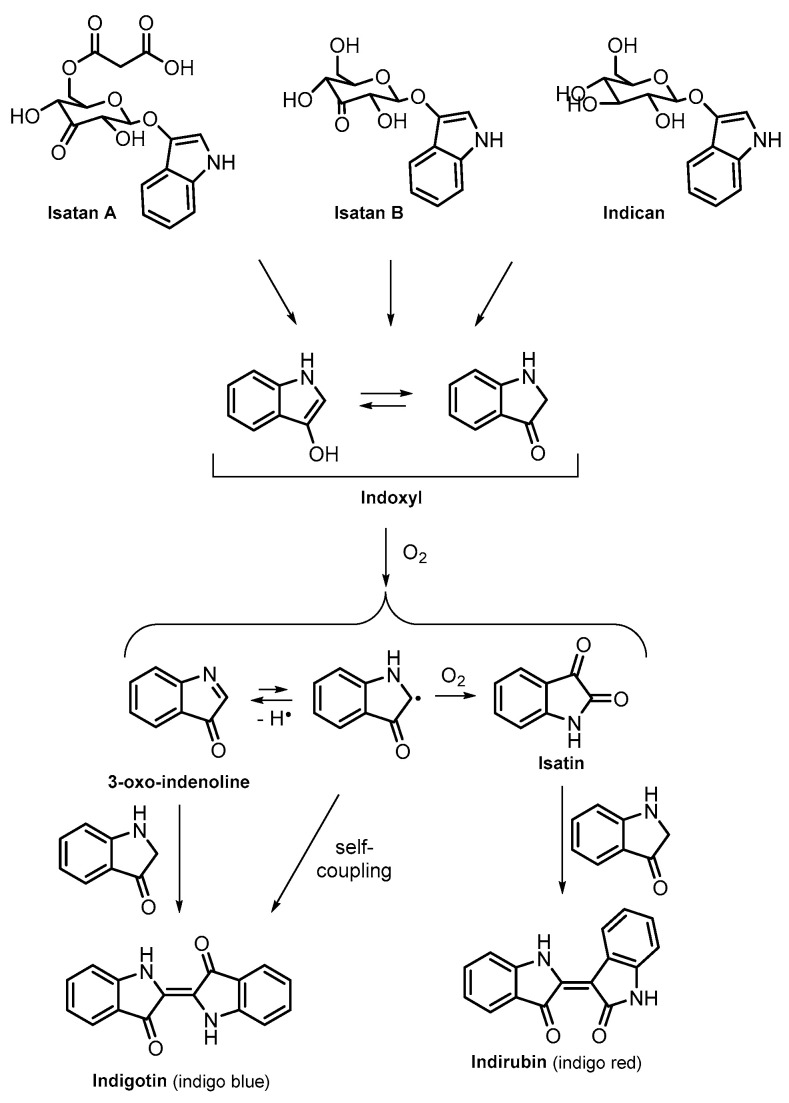
Indigotin and indirubin formation scheme.

**Figure 2 life-14-00059-f002:**
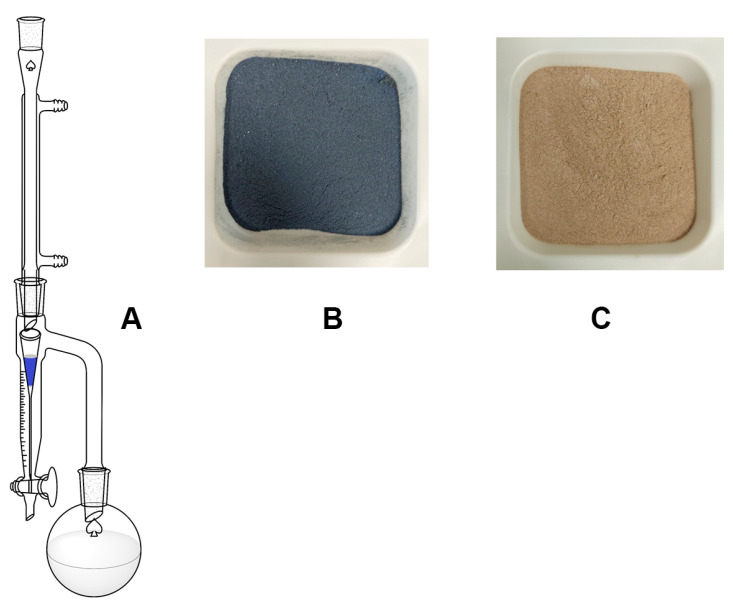
(**A**) Continuous extraction system. (**B**) Natural indigo composite before extraction. (**C**) Residue after complete extraction.

**Figure 3 life-14-00059-f003:**
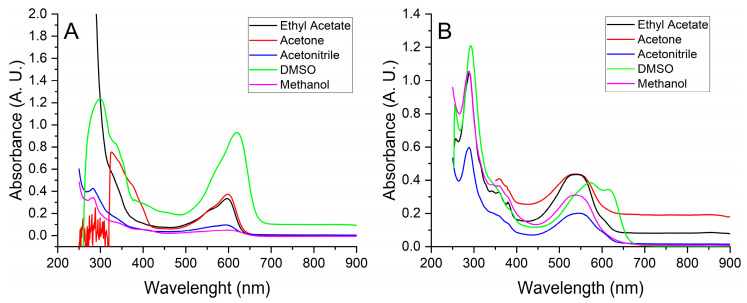
(**A**) UV-Visible absorption spectra of saturated solutions of synthetic indigo. (**B**) UV-Visible absorption spectra of saturated solutions of natural indigo (Product A).

**Figure 4 life-14-00059-f004:**
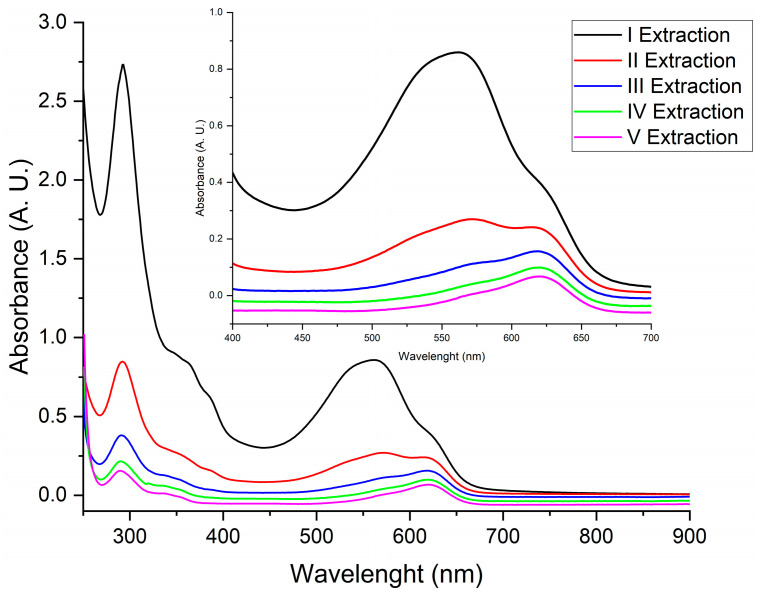
UV-Visible absorption spectra of sequential extracts with DMSO of Product B.

**Figure 5 life-14-00059-f005:**
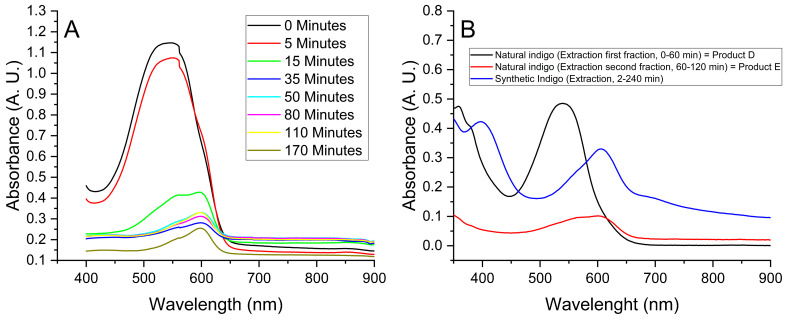
(**A**) UV-Vis spectra relative to the fraction collected during the continuous extraction of Product A with acetone. (**B**) UV-Vis spectra relative to the fraction collected during the continuous extraction of Product A or synthetic indigo with methanol.

**Figure 6 life-14-00059-f006:**
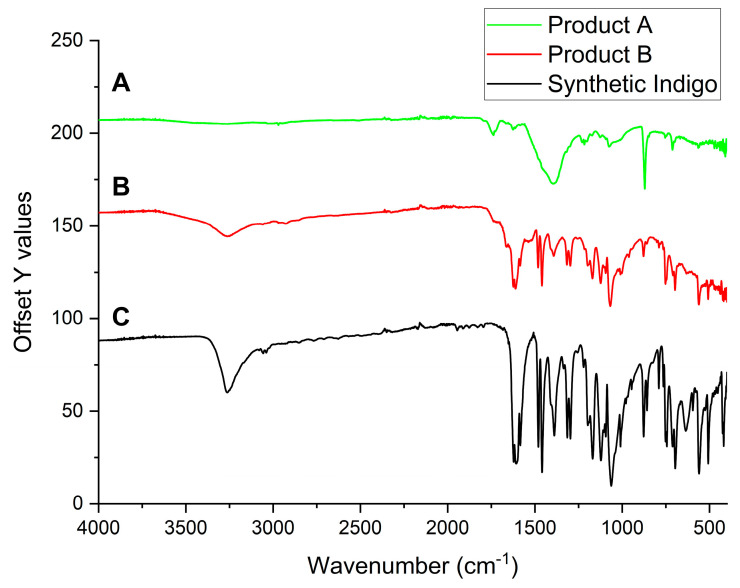
Percentage transmittance IR spectra carried out on samples of Product A (A), Product B (B), and synthetic indigo (C).

**Figure 7 life-14-00059-f007:**
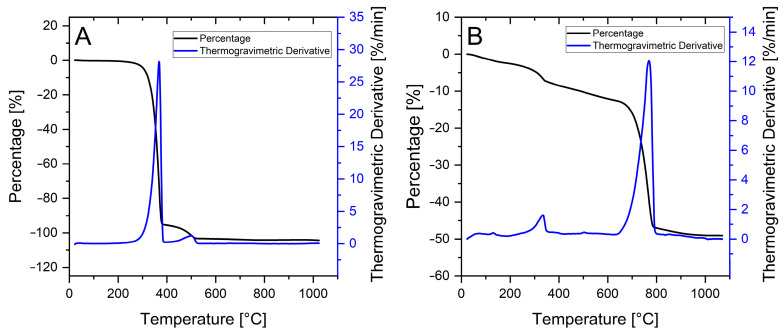
(**A**) TGA of synthetic indigo. (**B**) TGA of natural indigo (Product A).

**Figure 8 life-14-00059-f008:**
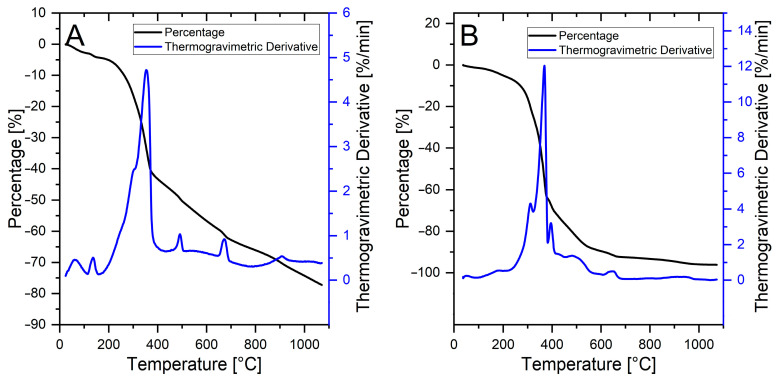
(**A**) TGA of carbonate-free natural indigo (Product B). (**B**) TGA of natural indigo extract (Product C) with acetone.

**Figure 9 life-14-00059-f009:**
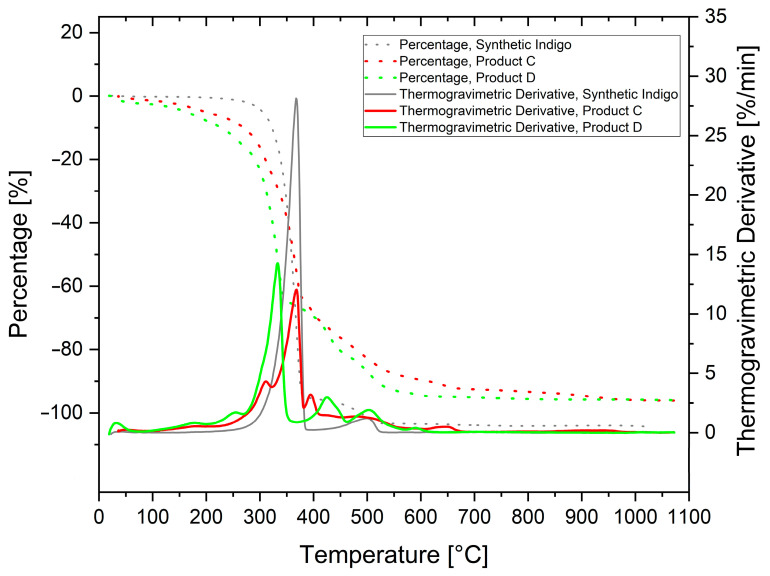
TGA–DTA of Product C (red line), Product D (green line), and synthetic indigo (grey line).

**Figure 10 life-14-00059-f010:**
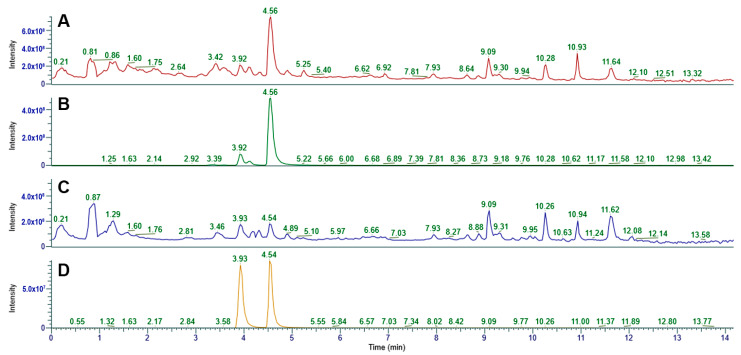
(**A**) Chromatogram relative to Product D, where the *y*-axis represents the Total Ion Current. (**B**) Chromatogram relative to Product D, where the *y*-axis represents the Single Ion Current relative to 263.08 *m*/*z*. (**C**) Chromatogram relative to Product E, where the *y*-axis represents the Total Ion Current. (**D**) Chromatogram relative to Product E, where the *y*-axis represents the Single Ion Current relative to 263.08 *m*/*z*.

**Table 1 life-14-00059-t001:** Relative amounts of extracted indigoids from Product A, using the continuous extraction process, after several times ^a^.

	Time (min)	15	60	180	240
Solvent					
acetone		20% ^b^	42%	100%	-
methanol		13%	20% ^b^	69%	100%

^a^ The first row is relative to the extraction with acetone; second row reports the extraction progress with methanol. ^b^ Fractionation at this time corresponds to the almost complete extraction of indirubin = Product D.

## Data Availability

The data are contained within the article.

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
