# Peer review of "A Continuous Extraction Protocol for the Characterisation of a Sustainably Produced Natural Indigo Pigment"

_life, 2023, doi:10.3390/life14010059_

Round 1

Reviewer 1 Report

Comments and Suggestions for Authors

In this work the authors developed a useful method to perform continuous extraction of indigo that results in a comprehensive extraction and fractionation of the pigment's components. The characterization data clearly show that the method can be employed for a facilitated determination of the components. 

Comments:

1) Since the authors presented the manuscript in the context of a sustainable experimental framework, all currently available sustainable production methods are supposed to be mentioned and indigoids are well-known synthetic enzymatic products: 10.1016/j.bcab.2022.102458, 10.1016/j.synbio.2023.06.006, 10.3390/ijms24032395, 10.1002/cctc.201901974, 10.3390/ijms232012544, 10.1186/s40643-023-00626-7.

2) A quantification of the amount of indigoids at different stages of the extraction is totally missing. The authors should add a table to support the results.

3) The abstract and the title should be re-written. The novelty of the work is clearly on the extraction method, whereas the title and the abstract point generically towards a characterization (method). It should be more straightforward. 

4) The conclusions should also be re-written to put more emphasis on the extraction. Key quantification data (see point 2) should also be included. The characterization part should only be in support of the extraction method.

Author Response

Please, check the attached pdf file

Reviewer 2 Report

Comments and Suggestions for Authors

(1) The yield of extraction process should be shown.

(2) For make a contrast well the absorption spectrum should normalized or the absorption coefficient as Y axis.

(3) The molecular weight should shown in is key to read the UHPLC-MS. 

Comments on the Quality of English Language

English very difficult to understand

Author Response

Please, check the attached pdf file.
